# Comparative Transcriptome and sRNAome Analyses Reveal the Regulatory Mechanisms of Fruit Ripening in a Spontaneous Early-Ripening Navel Orange Mutant and Its Wild Type

**DOI:** 10.3390/genes13101706

**Published:** 2022-09-22

**Authors:** Lanfang Mi, Dong Ma, Shuping Lv, Saibing Xu, Balian Zhong, Ting Peng, Dechun Liu, Yong Liu

**Affiliations:** 1College of Agronomy, Jiangxi Agricultural University, Nanchang 330045, China; 2College of Life Sciences, Gannan Normal University, Ganzhou 341000, China; 3National Navel Orange Engineering Research Center, Ganzhou 341000, China

**Keywords:** citrus mutation, fruit development and ripening, transcription factor, miRNA

## Abstract

A complex molecular regulatory network plays an important role in the development and ripening of fruits and leads to significant differences in apparent characteristics. Comparative transcriptome and sRNAome analyses were performed to reveal the regulatory mechanisms of fruit ripening in a spontaneous early-ripening navel orange mutant (‘Ganqi 4’, *C**itrus sinensis* L. Osbeck) and its wild type (‘Newhall’ navel orange) in this study. At the transcript level, a total of 10792 genes were found to be differentially expressed between MT and WT at the four fruit development stages by RNA-Seq. Additionally, a total of 441 differentially expressed miRNAs were found in the four periods, and some of them belong to 15 families. An integrative analysis of the transcriptome and sRNAome data revealed some factors that regulate the mechanisms of formation of early-ripening traits. First, secondary metabolic materials, especially endogenous hormones, carotenoids, cellulose and pectin, obviously changed during fruit ripening in MT and WT. Second, we found a large number of differentially expressed genes (*PP2C*, *SnRK*, *JAZ*, *ARF*, *PG*, and *PE*) involved in plant hormone signal transduction and starch and sucrose metabolism, which suggests the importance of these metabolic pathways during fruit ripening. Third, the expression patterns of several key miRNAs and their target genes during citrus fruit development and ripening stages were examined. csi-miR156, csi-miR160, csi-miR397, csi-miR3954, and miRN106 suppressed specific transcription factors (*SPLs*, *ARFs*, *NACs*, *LACs*, and *TCPs*) that are thought to be important regulators involved in citrus fruit development and ripening. In the present study, we analyzed ripening-related regulatory factors from multiple perspectives and provide new insights into the molecular mechanisms that operate in the early-ripening navel orange mutant ‘Ganqi 4’.

## 1. Introduction

The making of a fruit is a developmental process unique to plants. It requires a complex network of interacting genes and signaling pathways. Fleshy fruit involves three distinct stages, namely, fruit set, fruit development, and fruit ripening. Of these, ripening has received the most attention from geneticists and breeders, as this important process activates a whole set of biochemical pathways that make the fruit attractive, desirable, and edible for consumers. Changes in the citrus ripening period and color have been shown to attract the attention of consumers [1,2]. In recent years, research on citrus ripening regulation has increased slowly. Liu [3] found that ’Fengjie72-1′ and its spontaneous late-ripening sweet orange mutation ’Fengwan’ did not differ in DNA levels, but their main difference was at the transcriptional level. Transcription factors in the *ERF* (ethylene-responsive transcription factor) family were determined to be involved in hormone synthesis, fruit ripening, and carotenoid synthesis, etc. The *ERF* family also includes the most important transcription factors that regulate late ripening characteristics. Finally, the writer revealed the molecular mechanism of late ripening [4,5,6]. It was found that ‘Comune’ clementine and its late-ripening mutant ‘Tardivo’ (delayed fading and maturation) show significant differences in the transcriptional activity and ethylene signaling pathway of the *PSY* (phytoene synthase) gene [7]. Terol [8] found three clementine mature mutants: the early-ripening variety Arrufatina (ARR), intermediate-ripening variety Clemenules (CLE), and late-ripening variety Hernandina (HER). The MADS-box family of transcription factors are involved in the regulation of fruit ripening, and their expression levels were low in ARR and high in HER. The expression patterns of the ethylene synthesis genes encoding *SAM* (S-adenosyl methionine) and *ACC* (1-aminocyclopropane-1-carboxylate) synthase and *ACC* oxidase were opposite in early- and late-maturing mutants. Zhang [9] identified a number of metabolic pathways and key genes related to citrus ripening in Jincheng sweet orange and its late-ripening mutant.

In addition, citrus fruit color mutants are also important resources for studying fruit quality metabolism. Rodrigo [10] found a novel mutation (named Pinalate) in Navelate orange that produces distinctive yellow fruits instead of the typical bright orange coloration. Pinalate orange has a special pattern of linear carotene (phytoene, lycopene, zeta-carotene) accumulation in peel that is higher than the ring-structured and oxidized carotenoids upon fruit ripening. Research shows that Pinalate expresses a special bud mutation that results in material that is deficient in zeta-carotene desaturated enzymes and related factors; therefore, this specific material is important for research on carotenoid metabolism [11,12]. Guo [13] identified 12 transcription factors from the Guanxi pomelo mutant by RNA-Seq, three of which belong to the *MYB* (myeloblastosis) family and are mainly involved in the biosynthesis of anthocyanins. Transcription factors in the *NAC* (no apical meristem (NAM), *Arabidopsis* transcription activation factor (ATAF1/2), cup-shaped cotyledon (CUC2)) family play positive roles in the regulation of carotenoid accumulation. Xu comprehensively explained the gene expression and metabolic pathways of sweet orange (red-fleshed mutant) [14,15].

Overall, fruit growth and development, including maturation period, color and other qualities, are regulated by plant hormones, transcription factors, noncoding RNA and light. Wang [16] used ‘early red’ blood orange and ‘twenty-first century’ navel orange as materials to study the transcriptional characteristics during fruit development, and the results showed that some DEGs were involved in the metabolism of anthocyanin, sucrose and citric acid during the fruit ripening stage. Lu [17] found that the *CsMADS6* transcription factor in citrus can positively regulate carotenoid metabolism by directly regulating the expression of *LCYB1* (lycopene β cyclase 1), *PSY*, *PDS* (phytoene desaturase), *CCD1* (carotenoid cleavage dioxygenases 1) and other carotenoid genes.

Previous studies [18] have found that ethylene has a positive regulatory effect on the coloration of pepper fruits, but ABA inhibits fruit coloration. These results show that ABA plays a certain role in the degreening process. This study provides new insight into the regulatory mechanisms of ABA and ETH in fleshy fruit ripening and provides a new approach for the study of ABA and ETH changes in the carotenoid biosynthesis and chlorophyll degradation processes. Song [19] found that ABA is involved in regulation of fruit ripening in strawberry. Induction by ABA positively regulated the K+ channel amino acid gene *FaKAT1* and promoted fruit coloration and ripening; in contrast, the expression of the *FaKAT1* gene was downregulated when the ABA levels were reduced, so they speculated that *FaKAT1* may be the target gene of the ABA signaling pathway. As well as, *CsNCED1* might be a key gene in the biosynthesis of ABA; thus, *NCED1* (9-cis-epoxycarotenoid dioxygenase) was the significant regulator for orange fruit ripening [4].

Small RNAs are 20–24 nt noncoding RNA molecules that are mainly involved in posttranscriptional regulation [20]. The known miRNAs involved in the regulation of fruit ripening in horticultural plants include miR156/*SPL* (ovary development), miR160/*AFR* (fruit shape), miR164/*NAC* (fruit ripening), miR167/*ARF* (fruit initiation), miR172/*AP2* (fruit size), miR390/TAS3/*ARF* (fruit formation), miR393/*TIR1/ARB2* (fruit set), miR397/*LAC* (fruit quality), miR399/*PHO2* (fruit quality), miR828/miR858/*MYB* (coloration), and *Fan*-miR73/*ABI5* (fruit ripening).

A total of 183 known miRNAs and 38 novel miRNAs have been identified from leaves, flowers and fruits of sweet orange in previous research. It has been demonstrated that csi-miR164 is highly expressed at the fruit ripening stage and interacts with target gene NAC transcription factors [21].

Drawing on insights from previous research and based on the molecular biology research platform, we conducted an investigation of ‘Ganqi 4’, which matures early and peels rapidly, at the physiological and molecular levels. Then, we identified the key genes and pathways that can regulate fruit development at the transcriptional and posttranscriptional levels to reveal the molecular mechanisms involved.

## 2. Materials and Methods

### 2.1. Plant Material and Treatments

The WT ‘Newhall’ navel orange (*C. sinensis* L. Osbeck) and its spontaneous early-ripening mutant (MT) ‘Ganqi 4’ were both cultivated in the same orchard located in Longnan, Ganzhou City, China (N 24°84′, E 114°68′,). We collected fruit samples from 70 to 205 days after flowering (DAF), which were from three different trees of same genotype every 15 days in 2018, and fruit without visible disease and mechanical wounding were selected for the study. Fruit samples with large differences in peel color were harvested at 100 (no obvious difference in color), 130 (MT begins to fade), 175 (MT mature, WT immature), and 205 DAF (MT full ripeness) for quantification of metabolic materials and RNA-Seq. Twelve representative fruits were sampled from each tree, and all the fruits were divided into three replicates. After wiping each peel and separating the pulp and the peel, the peel was sliced. Combining the sliced WT peel samples with one another (as for the MT samples), they were frozen rapidly in liquid nitrogen and maintained at −80 °C. A portion of each combined sample was used for extracting the RNA. Another aliquot was used for transcriptome sequencing and sRNA sequencing. A portion of each combined sample was used for the determination of carotenoid, cellulose, pectin and endogenous hormone composition and concentration. The remaining portions were frozen at −80 °C until use.

### 2.2. Quantification of Carotenoid, Cellulose, Pectin Substances and Related Enzyme Activities and Endogenous Hormones

According to a method described previously [22], 1 g freeze-dried peel powder material was accurately used for extracting carotenoids. Each sample has three repetitions, and a high-performance liquid chromatography (HPLC) method was used to determine the carotenoid composition and concentrations in the WT and MT fruit peels harvested at 100, 130, 175, and 205 DAF.

Extraction and quantitative analysis of pectin was performed using the carbazole colorimetric method according to Taylor and Liu [23,24]. The method of determining cellulose and hemicellulose content was slightly modified from a method described previously [25]. We used a method described previously to extract the enzyme solution [26]. The activities of PG (polygalacturonase) and CX (cellulase) were determined by the DNS colorimetric method [27], and that of PME (pectin methylesterase) was determined by the NaOH titration method [28].

We used an enzyme-linked immunosorbent assay (ELISA) [29,30,31] to determine the endogenous hormone (IAA, GA, CTK, ABA, ETH, JA, SA) contents in peel samples, and the specific operations were carried out according to the instructions of individual ELISA kit from Shanghai Heng Yuan Biological Technology Company. Each sample was assayed with three replicates.

### 2.3. Library Preparation for Transcriptome Sequencing

RNA extraction method was performed as described previously [22]. The WT and MT fruit peels were subjected to RNA-Seq using an Illumina PE150 at Beijing Novogene Co., Ltd. (Tianjin) in 2009. Briefly, 3 µg of total RNA from each sample was used for library construction. Sequencing libraries were generated using the NEBNext^®^ Ultra TM RNA Library Prep Kit for Illumina^®^ (New England Biolabs, MA, USA) following the manufacturer’s protocol. To preferentially select cDNA fragments 150~200 bp in length, the library fragments were purified with an AMPure XP system (Beckman Coulter, Beverly, NJ, USA). PCR products were purified (AMPure XP system), and library quality was assessed on the Agilent Bioanalyzer 2100 system. The clustering of the index-coded samples was performed on a cBot Cluster Generation System using TruSeq PE Cluster Kit v3-cBot-HS (Illumina) according to the manufacturer’s instructions. After cluster generation, the library preparations were sequenced on an Illumina HiSeq platform, and 125 bp/150 bp paired-end reads were generated.

### 2.4. Library Preparation for Small RNA Sequencing

As with transcriptome sequencing, 3 μg of total RNA was used for small RNA library construction. Sequencing libraries were generated using NEBNext^®^ Multiplex Small RNA Library Prep Set for Illumina^®^ (New England Biolabs, MA, USA.) according to the manufacturer’s protocol. Briefly, NEB 3′ SR Adaptor was directly and specifically ligated to the 3′ end of miRNA, siRNA and piRNA. After the 3′ ligation reaction, the SR RT Primer hybridized to the excess 3′ SR Adaptor (that remained free after the 3′ ligation reaction) and converted the single-stranded DNA adaptor into a double-stranded DNA molecule. This step is important to prevent adaptor-dimer formation. In addition, dsDNAs are not substrates for ligation mediated by T4 RNA ligase 1 and therefore do not ligate to the 5′ SR Adaptor in the subsequent ligation step. The 5′ end Adaptor was ligated to the 5′ ends of miRNAs, siRNA and piRNA. Then, first strand cDNA was synthesized using M-MuLV Reverse Transcriptase (RNase H-). PCR amplification was performed using LongAmp Taq 2X Master Mix, SR Primer for Illumina and index (X) primer. PCR products were purified on an 8% polyacrylamide gel (100 V, 80 min). DNA fragments corresponding to 140~160 bp (the length of small noncoding RNA plus the 3′ and 5′ Adaptors) were recovered and dissolved in 8 μL elution buffer. Finally, library quality was assessed on the Agilent Bioanalyzer 2100 system using DNA High Sensitivity Chips. The clustering of the index-coded samples was performed on a cBot Cluster Generation System using TruSeq SR Cluster Kit v3-cBot-HS (Illumina) according to the manufacturer’s instructions. After cluster generation, the library preparations were sequenced on an Illumina HiSeq 2500/2000 platform, and 50 bp single-end reads were generated.

### 2.5. Data Analysis

The data analysis process mainly includes the following steps, such as quality control, reads mapping to the reference genome, quantification of gene expression level, differential expression analysis, and Gene Ontology (GO) and Kyoto Encyclopedia of Genes and Genomes (KEGG) enrichment analysis of differentially expressed genes. Clean reads for subsequent analysis were obtained after filtering the original data and checking the sequencing error rate, the distribution of GC content and Q20/Q30 (The percentage of bases with Phred values greater than 20 and 30 in the total base, when they were greater than 85%, that the data quality is qualified). Genome sequence of *C. sinensis* annotated (http://citrus.hzau.edu.cn/index.php (accessed on 25 July 2022)) was used as reference. HTSeq v0.6.0 was used to count the read numbers mapped to each gene, and FPKM (fragments per kilobase of transcript sequence per million) of each gene was calculated to represent the level of gene expression. The edgeR R package (3.12.1) was used to identify the differentially expressed genes (DEGs) based on the following criteria: padj < 0.05 and |log2 ratio| ≥ 1. GO enrichment analysis of DEGS was implemented by the clusterProfiler R package, and we used clusterProfiler to test the statistical enrichment of DEGs in KEGG pathways (https://www.genome.jp/kegg/ (accessed on 15 October 2021)).

All sRNAome raw data were processed through custom perl and python scripts, clean reads were obtained by removing reads containing ploy-N, with 5′ adapter contaminants, without 3′adapter or the insert tag, containing ploy A or T or G or C and low quality reads from raw data. Then, the small RNA tags (the range of length was 18–30 nt) were mapped to reference sequence by Bowtie (0.12.9) without mismatch to analyze their expression and distribution. MiRBase20.0 (https://www.mirbase.org/ (accessed on 20 December 2021)) was used as reference to obtain the potential miRNA and draw the secondary structures. Combined with the characteristics of hairpin structure of miRNA precursor, we integrated the software of miREvo (v1.1, State Key Laboratory of Biocontrol, Sun Yat-sen University, Guangzhou, China) and mirDEEP2 (v2_0_0_5, The Helmholtz Association of National Research Center, Berlin, Germany) to predict novel miRNA, and we used psRobot_tar in psRobot to predict the target genes of miRNAs. Differential expression analysis was performed using the DEGseq (2010) R package, q-value < 0.05 and |log2 ratio| ≥ 1 was set as the threshold for significantly differential expression by default.

### 2.6. RNA Extraction and qRT-PCR Analysis

To verify the expression accuracy of genes that were correlated with important metabolic pathways, 26 selected DEGs and 8 selected differentially expressed miRNAs were analyzed by qRT-PCR.

26 DEGs: A total of 1 µL of RNA was reverse transcribed for first-strand cDNA synthesis using the first strand cDNA synthesis kit (Simgen) according to the manufacturer’s protocol. Gene-primer pairs were designed with Primer Premier 5 software and then used for qRT-PCR verification. PCR was performed using a SYBR Green PCR Mix kit (Simgen) with 10-µL reaction systems on a ROCHE LC96 Real-Time PCR instrument to detect the gene expression levels, and the relative quantification was calculated by the 2^−^^ΔΔCT^ method. The specific primers are listed in Appendix A.

Eight differentially expressed miRNAs: Peel miRNAs were extracted using miRNA purification kits (Simgen, Hangzhou, China), and 2 µg of miRNA was reverse transcribed for first-strand cDNA synthesis using a miRNA First-Strand cDNA Synthesis kit (A&D Technology Corporation) according to the manufacturer’s protocol. The primers were designed with Primer Premier 5 software. PCR was performed using a stem-loop qRT-PCR Assay kit (A&D Technology Corporation). The specific primers are listed in Appendix A.

According to the quantitative method of Wu [4], we carried out differential expression verification of target genes. qRT-PCR was performed with three biological replicates, and the primers are listed in Appendix A.

### 2.7. Statistical Analyses

Values presented are the means ± standard deviations (SD) of three replicates. Statistical analyses were carried out by analysis of variance (ANOVA) using SAS (SAS Institute, Cary, NC, USA) software.

## 3. Results

### 3.1. Changes in Metabolic Materials during Fruit Ripening in MT and WT

With ripening, the physiological and biochemical qualities of citrus fruit undergo great changes. For example, the skin is degreened and develops coloration, and the peel becomes relaxed. All these processes are caused by changes in the contents of secondary metabolites in fruits. In this study, we dynamically detected the metabolites in the peel and determined the trends in levels of endogenous hormones, carotenoids, cellulose and pectin during ripening of MT and WT fruits (Figure 1E).

The endogenous hormones in fruits mainly include five categories, such as IAA, GA, CTK, ABA and ETH. In addition, SA and JA also play important roles in the fruit ripening process. The dynamic changes in these hormones during fruit development of MT and WT are presented in Figure 1A, which shows that the IAA content appeared to increase from 100 DAF and then appeared to rapidly decline from 175 DAF; on the whole, the levels of IAA were significantly higher in MT than in WT. At 175 DAF, the GA content was the highest and was significantly different in the fruit peel of MT and WT and then decreased rapidly after 175 DAF, and the content of GA in MT was significantly higher than that in WT before fruit degreening (Figure 1B). The patterns of change in CTK content were identical in MT and WT (Figure 1C). Additionally, the trend in ABA was similar; the ABA content increased with fruit ripening, and the content in MT was higher than that in WT at 100 DAF and 175 DAF (Figure 1D). In the four developmental stages, the ETH content in WT was significantly higher than that in MT, and both increased first and then decreased (Figure 1E). Patterns for JA and SA were the same in MT and WT; the content first decreased but then increased, and it was significantly higher in MT than WT at 175 days (Figure 1F,G).

The pigment that accumulated in citrus peel was mainly carotenoids. In this study, we found that the peel of MT was orange-yellow and that of WT was orange-red when they were ripening; therefore, we analyzed the reason for the color difference through the accumulation characteristics of different carotenoids (Figure 1P,R). The content of β-carotene in WT was significantly higher than that in MT from degreening to ripening, and the pattern of change in lutein was similar to that of β-carotenoids. At 100 DAF, the content of zeaxanthin in MT was significantly higher than that in WT, and with fruit ripening, the content showed a rapid upward trend, but the content in WT was significantly higher than that in MT (Figure 1Q). Unfortunately, lycopenes were not detected in the two varieties.

Dynamic analysis of cellulose, pectin and related enzyme activity in peel is shown in Figure 1H–O. During fruit development, the cellulose and hemicellulose contents in MT were significantly lower than those in WT (Figure 1H,I); in contrast, CX was significantly higher in MT (Figure 1M). The contents of total pectin and raw pectin in MT were significantly lower than those in WT, but the content of soluble pectin showed the opposite pattern (Figure 1J–L). The activity of PG in MT was significantly higher than that in WT, but there was no difference in PME activity (Figure 1N,O).

### 3.2. Fruit Transcriptome Differences between MT and WT during Fruit Ripening

#### 3.2.1. Expression Analysis of Different Genes in MT and WT

We used transcriptome sequencing in this research and performed a differential expression analysis of the two genotypes MT and WT according to the results. We have obtained 2.31 billion raw reads and 2.28 billion clean reads from all the samples; the error rate was below 0.03%, and the Q30 was higher than 91%. The proportion of clean reads in each sample was more than 98%, and they will be used for the subsequent data analysis. EdgeR (applying the RSEM quantitative approach) software was used to determine significant differences in expression. The results showed that a total of 10,792 genes were differentially expressed between MT and WT at the four fruit developmental stages (Figure 2B). Of these, 2633 were at 100 DAF, 3038 were at 130 DAF, 2804 were at 175 DAF, and 2317 were at 205 DAF, as detailed in Appendix A. The number of upregulated genes was less than that of downregulated genes at various stages, and the number of DEGs first increased and then decreased. The largest number appeared at 130 DAF, followed by 175 DAF (Figure 2A).

#### 3.2.2. Annotation of DEGs in MT and WT

To explore the biological functions of the coordinated expression of different genes in the two varieties, the goseq package was used in this study to conduct GO enrichment analysis of all DEGs. As shown in Figure 2C, according to the results from GO term enrichment, the number of genes with particular functions varied among developmental stages of the fruit. Biological processes (1598 genes at 100 DAF, 1934 genes at 130 DAF, 1774 genes at 175 DAF and 1457 genes at 205 DAF) and metabolic processes (1246, 1539, 1407, and 1156, respectively) were the major categories annotated under biological processes, followed by single-organism metabolic processes (458, 536, 498, and 420) and oxidation–reduction processes (309, 349, 316, and 287, respectively). Catalytic activity (1196, 1459, 1317, and 1117), transferase activity (487, 599, 550, and 456) and oxidoreductase activity (314, 350, 319, and 305, respectively) were the major categories annotated as molecular function. According to the GO terms, we found that the DEGs that were related to cellular components were mostly found at 130 and 175 DAF and were less prevalent at 100 and 205 DAF, including thylakoid (0, 49, 38, and 0), photosystem (0, 47, 33, and 0) and photosynthetic membrane (0, 48, 36, and 0, respectively) DEGs, as detailed in Appendix A.

We used KOBAS (2.0) to perform pathway enrichment analysis. KEGG analysis divided the differentially expressed genes into 56 metabolic pathways that contained five or more differentially expressed genes, as shown in Appendix A. The major pathways related to metabolism in MT and WT are listed in Table 1. We found that there were many genes involved in the metabolic pathway (more than 250) and biosynthesis of secondary metabolites (over 150) during fruit ripening. In addition, the metabolic pathways that were closely related to the formation of fruit quality were plant hormone signal transduction, starch and sucrose metabolism, biosynthesis of amino acids, sesquiterpenoid and triterpenoid biosynthesis, carotenoid biosynthesis and flavonoid biosynthesis. The plant hormone signal transduction pathway exhibited 43 enriched differentially expressed genes at 175 DAF, the starch and sucrose metabolism pathway showed 34 enriched DEGs at 100 DAF, and the carotenoid biosynthesis pathway displayed 9 enriched DEGs at 100 DAF.

In this study, MT is a spontaneous early-ripening navel orange mutant, so endogenous hormones play an important role in fruit ripening. After analysis, we obtained 38 genes involved in plant hormone metabolism, as shown in Table 2, including five genes related to ABA signal transduction (*Cs5g34270*, *Cs5g07700*, *Cs1g23060*, *Cs8g19140*, and *Cs4g14980*) and nine genes involved in ethylene synthesis and signal transduction (*Cs5g03060*, *orange1.1t00416*, *Cs2g02500*, *Cs5g20590*, *Cs9g08850*, *orange1.1t02185*, *Cs4g17960*, *Cs5g29870*, and *Cs9g10650*). Many genes also affected the JA metabolism pathway, approximately eighteen, such as *JAZ* (*orange1.1t00464*, *Cs1g17220*, *Cs4g06520*, *Cs7g02820*, *Cs4g07130*, and *Cs1g17210*), *MYC2* (*orange1.1t00550*, *orange1.1t01021*, and *Cs5g01450*), *LOX* (*orange1.1t03769*, *orange1.1t04376*, *Cs1g17380*, and *orange1.1t03771*), *AOS* (*Cs3g24230*), and *OPR* (*orange1.1t03727*, *Cs5g17900*, *Cs4g15220*, and *Cs5g17880*). Two genes affected the IAA signaling pathway, *AUX1* (*Cs8g02900* and *Cs7g31320*). Most of these genes were downregulated in MT.

The fruit of MT is easy to peel compared with WT fruit, so we also found that 15 genes were involved in the starch and sucrose metabolism pathways, such as *PG* (*orange1.1t02530* and *orange1.1t02529*), which were upregulated significantly, and the majority of *SPS* (*Cs7g05690*, *orange1.1t03668*, *Cs4g05380*, and *Cs5g19060*) genes were upregulated in MT, but *PEs* (*Cs4g06710*, *Cs5g33420*, and *Cs2g17070*) displayed a mixed expression pattern.

MT has a peel color different from that of WT, that is, orange-yellow peel; therefore, Table 2 shows that 8 genes were found to be involved in the carotenoid biosynthesis pathway, including *orange1.1t02108*, *Cs6g15910*, *Cs3g11040*, *Cs4g14850*, *Cs9g19270*, *Cs5g14370*, *Cs9g11260*, and *Cs2g03270*. Some of the genes were upregulated, and the remainder were downregulated.

#### 3.2.3. Verification of Differentially Expressed Genes

To confirm whether the expression patterns of genes related to ripening were consistent with the transcriptional sequencing data, approximately 50 genes were subjected to quantitative analysis by qRT-PCR. We designed gene-specific primers and constructed a cDNA library, and some of the quantitative results are shown in Figure 3. Compared with the RNA-Seq results, the expression levels of these genes were consistent by qRT-PCR.

#### 3.2.4. Expression Pattern Comparisons for ABA, JA Synthesis, Signal Transduction and Some Starch and Sucrose Metabolism-Related Genes during Fruit Development and Ripening in MT and WT

In this research, we found that some of the genes played important roles in quality formation and ripening regulation at six developmental stages (100, 130, 160, 175, 190, and 215 DAF), including 10 ABA-related genes, 10 JA-related genes and 5 peel structure-related genes, as shown in Figure 4. In the ABA biosynthesis pathway, *CsVDE* (violaxanthin de-epoxidase), *CsCHYB*, and *CsNCEDs* played crucial roles, especially *CsNCEDs* with fruit ripening. Their expression levels increased gradually during fruit development and increased rapidly in later developmental stages in MT, so the ABA content in MT accumulated earlier than that in WT. In the ABA signal transduction pathway, *PP2C* and *SnRK2* were detected, and they played an important role in regulating citrus fruit ripening as the core transmitter ABA-PYR-PP2C-SNRK2. At the early stage of fruit development, *PP2C* was at a low level in MT, but it increased at the late stage. *SnRK2* had similar expression patterns at the early fruit development stage in MT and WT, but the expression level in MT was higher than that in WT at the late stage. The results showed that ABA accumulated earlier in MT than in WT and with lower expression of the inhibitory factor *PP2C*; therefore, ABA regulated fruit ripening earlier in MT. However, *SnRK2* activity would still be affected even if ABA accumulation increased due to the significant increase in *PP2C* expression levels with fruit ripening, so the regulation of ABA in fruits was not obvious at the late stage of fruit development.

In the JA biosynthesis and signal transduction pathway, the three transcription factors *OPR*, *MYC2* and *JAZ* are very important. *OPR* is a key enzyme in the JA synthesis pathway. In this study, we found that the expression of the *OPR* gene was upregulated at 160 DAF and reached the highest expression from 160 DAF to 175 DAF in MT, while the highest expression level of the *OPR* gene appeared after 175 DAF in WT. The results indicated that the peak of *OPR* appeared earlier in MT than in WT, so JA accumulated earlier in MT. As a negative regulator of *JAZ* in JA signal transduction, the level of *JAZ* expression gradually increased, and the expression pattern was similar to that during fruit development and ripening, but the expression level was higher in WT than in MT. The expression level of *MYC2* was higher in WT at 175 DAF, but there were no differences at other stages. DEG analysis showed that JA was synthesized and accumulated in the early stage in MT and regulates fruit development through signal transduction; compared with MT, the accumulation and regulation of JA occurred later in WT.

In the starch and sugar metabolism pathway, α-glucosidase and β-fructofuranosidase are the key enzymes in fructose synthesis, and their expression levels were upregulated during fruit development and increased with fruit ripening. The expression levels of *PG* and *GUAT* in MT showed an obvious upward trend with fruit ripening, and the expression levels of PE were already high at the early stage of fruit development. Comprehensive analysis showed that sugar metabolism appeared earlier and that pectin decomposed earlier and faster in MT than in WT. Therefore, this was the main reason for easy peeling in this variety.

### 3.3. Differential Expression Analysis of miRNA in MT and WT

Clean reads that were used for further analysis were obtained from raw reads by removing low-quality reads and reads containing joints. sRNAs within a certain length range were screened from clean reads, which were assigned to the reference sequence by Bowtie, and the reads aligned to the reference sequence were compared with the specified range sequence in miRBase to obtain detailed information on matched sRNAs, including secondary structure, sequence, length and quantity of known and unknown miRNAs, as shown in Table 3. Totals of 14,316,231; 15,465,813; 16,418,922; and 15,237,868 clean reads were generated from MT at 100, 130, 175, and 205 DAF, respectively. Totals of 16,423,940; 16,986,556; 14,755,090; and 16,507,044 were generated from WT at different stages of fruit development, including 1,518,461; 1,879,119; 818,850; and 2,202,926 unique reads in MT and millions of unique reads in WT (2,381,531; 1,736,884; 1,068,268; and 1,729,613, respectively). In the mapped sRNAs in MT, the percentages of known and novel miRNAs were 2.01% and 0.11% at 100 DAF, 2.77% and 0.14% at 130 DAF, 2.16% and 0.07% at 175 DAF, and 2.96% and 0.17% at 205 DAF, respectively. In WT, the percentages of known and novel miRNAs were different at 100 DAF (3.66% and 0.17%, respectively), 130 DAF (3.20% and 0.14%, respectively), 175 DAF (1.32% and 0.16%, respectively), and 205 DAF (3.95% and 0.23%, respectively). We found that the sequence lengths ranged between 18 and 30 nt (Table 3), and the most common sRNA length was 24 nt, followed by 21 nt, as shown in Figure 5C.

#### 3.3.1. Expression Analysis of Known and Novel miRNAs in MT and WT

A total of 441 differentially expressed miRNAs were found in the four periods (including the miRNAs that reappeared in four stages), some of which belong to 15 families (Appendix A). The number of upregulated miRNAs was 203, which was 35 less than that of downregulated miRNAs, and except for at 130 DAF, there were more downregulated than upregulated miRNAs (Figure 5A,B). A total of 116 novel miRNAs were identified, and the number of downregulated miRNAs was 86, which was much higher than the number of upregulated miRNAs, as shown in Figure 5A. At 175 DAF, the number of downregulated miRNAs was the largest, followed by 100 DAF, and there was no significant difference in number between 135 and 205 DAF. All of the precursors of miRNAs had regular stem–loop secondary structures (Appendix A), including known miRNAs and novel miRNAs. In this study, we set q-value <0.05 and |log_2_ ratio| ≥ 1 as the criteria for significant differences.

To show the numbers of common and unique miRNAs differentially expressed between MT and WT in the four different periods, we constructed a Venn diagram. The large oval represents the combination of each different period, the sum of the numbers in each oval represents the total number of different miRNAs, and overlapping represents the number of differences among the combinations. As shown in Figure 6B, a total of 21 different miRNAs were found in the four periods: 12 specific miRNAs were detected at 100 DAF, 17 specific miRNAs were detected at 130 DAF, 39 specific miRNAs were detected at 175 DAF, and 11 specific miRNAs were detected at 205 DAF. With the development of fruit, the influence of miRNAs on fruit ripening gradually increased, and the maximum difference appeared at 175 DAF.

Most of the major known miRNAs and novel miRNAs differentially expressed between MT and WT were downregulated, some miRNAs were up- or downregulated, and only one miRNA was upregulated at different stages of fruit development. The expression of csi-miRNA159b-5p and csi-miRNA3951b-5p was downregulated in all four periods, and the expression levels were high. The novel miRNAs novel_106 and novel_127 were also downregulated, and both had high expression (Figure 6A).

#### 3.3.2. Prediction and Annotation of Target Genes

In this study, the target genes were predicted by TargetFinder software. From the sRNAome data, we found 213 specific miRNAs and obtained 32,609 miRNA/target pairs. According to the GO enrichment analysis, the target genes were classified as biological process, molecular function and cellular component genes. Appendix A shows that the functions of most of the target genes belong to the biological process and molecular function categories. At the four fruit development stages, the functions of the enriched target genes were different; the maximum number appeared at 175 DAF, the second highest number appeared at 205 DAF, and the last two were 130 DAF and 100 DAF. Target genes were predicted based on scores and annotation. The smaller scores are better. In Appendix A, we listed the common different miRNAs, specific different miRNAs, and their target genes in the four developmental stages. Among the commonly known miRNAs, csi-miR156f-5p, csi-miR399f-5p and csi-miR477e-3p each have 8 target genes; csi-miR160c-5p, csi-miR3951b-5p, csi-miR399b-3p, csi-miR530b-5p and csi-miR827 are each in the range of 4–6 for the number of genes targeted, and the others have 1–2 target genes. Among the common novel miRNAs, miRN106 has 10 target genes, and miRN68 has 6. Among the specific miRNAs, csi-miR477a-5p, csi-miR156g-3p and csi-miR399b-5p had high differential expression at 100 DAF, and target genes were predicted to be 1, 7, and 3, respectively. At 130 DAF, the miRNA/target gene pairs were csi-miR399b-5p (*Cs3g22340*, *Cs7g02280*, *Cs5g22690*), csi-miR167d-3p (*orange1.1t03541*), csi-miR398a-5p (*Cs5g05050*), and csi-miR172b-5p (*Cs7g06850*, *orange1.1t05187*, *Cs9g02960*). At 175 DAF, five novel miRNAs had high expression, miRN41, miRN73, miRN54, miRN135 and miRN58. At 205 DAF, the differentially expressed miRNA was csi-miR3627b-3p, and the target gene was *Cs4g04300*.

#### 3.3.3. Functional Analysis of Target Genes in Which Major miRNAs Were Involved in Fruit Development

In this study, a total of 44 target genes of major miRNAs involved in fruit development were identified, as shown in Table 4. miR156f-5p was upregulated and downregulated in MT vs. WT, and the target genes of miR156f were eight *SPL* genes, including *SP13B* (*orange1.1t02597*), *SPL16* (*Cs7g10990*), *SPL7* (*Cs1g03640*), *SPB1* (*Cs2g05730*), *SPL9* (*orange1.1t02265*), *SPB2* (*Cs2g23550*), *SPL6* (*Cs7g11770*), and *SPL2* (*Cs7g10830*), which are important for regulating fruit growth and development. miR397 was strongly upregulated at 175 DAF and 205 DAF, and it had 14 *LAC* target genes, for example, *LAC11* (*Cs6g11860*), *LAC4* (*Cs6g07800*), *LAC4* (*Cs8g19850*), *LAC7* (*Cs6g07410*), *LAC17* (*Cs7g23490*), *LAC7* (*Cs6g07400*), *LAC7* (*Cs6g07450*), *LAC17* (*Cs8g18800*), *LAC17* (*Cs6g06920*), *LAC17* (*Cs6g06880*), *LAC17* (*Cs8g17630*), *LAC17* (*Cs8g17350*), *LAC17* (*Cs6g06890*), and *LAC22* (*Cs7g31620*), which regulate fruit quality. The target genes of miR160c are four *ARF* genes, which are important for regulating fruit size. We also obtained four target genes (*AP2*/*RAP*) of miR172a and seven target genes (*NAC*) of miR3954, which regulate fruit growth and development, abiotic stresses, etc. Among the novel miRNAs, a total of five target genes of miR106 included *TCP2* (*Cs2g08080*), *TCP4* (*Cs2g15820*), *TCP4* (*Cs7g25460*), *MYB33* (*Cs3g06390*) and *NAC29* (*Cs8g16870*), which are transcriptional regulators involved in development and stress responses, as detailed in Appendix A.

#### 3.3.4. Comparative Expression Patterns of miRNAs and Target Genes in MT and WT

According to the sRNA-Seq results, we analyzed the expression patterns of eight miRNAs (five known miRNAs and three novel miRNAs) at six developmental periods (100, 130, 160, 175, 190, and 205 DAF). The results showed that the expression pattern of miR156 was similar between the two varieties and that the expression levels were low. The expression pattern of miR397 in WT was upregulated first during the development of WT fruit and then decreased, and it reached the highest expression level at 175 DAF. Compared with WT, the expression level of MT increased slowly during the early stage of fruit development and rapidly increased in the mature stage. In MT and WT, miR3954 and miR172 shared a similar pattern, both first increasing and then declining; the maximum level of miR3954 in MT appeared at 190 DAF and in WT at 175 DAF. The highest expression level of miR160 in MT was four times that in WT at 160 DAF. The expression pattern of miRN127 in MT was opposite to that in WT; the level of miRN127 was the highest at 100 DAF and 190 DAF in MT and showed a down-up-down pattern, while the lowest level in WT was at 190 DAF. The expression levels of miRN127 in other stages were lower than those in MT except for at 175 DAF. miRN54 was present at lower levels in MT and WT, but it was eight times higher in MT than in WT at 160 DAF.

To verify the regulatory functions of miRNAs, we performed qRT-PCR analysis on the selected target genes of eight candidate miRNAs involved in fruit development, as shown in Figure 7. The quantitative analysis of 29 target genes showed that the expression patterns of most target genes were different in MT and WT (Figure 8 and Figure 9), but eight target genes showed similar expression patterns, including *SP13B* (*orange1.1T02597*), *SPL9* (*orange1.1T02265*), *SPL2* (*Cs7g10830*), *PP323* (*orange1.1T03355*) and *HSR4* (*Cs1g19320*). Most of the target genes were negatively regulated by miRNAs; for example, miR156f negatively regulated the *SPL* gene, and *NAC* genes were negatively regulated by *miR3954*. The results described above indicated that miRNAs regulated target genes, namely, transcription factors, which affected fruit development and ripening.

## 4. Discussion

Fruit ripening involves the well-orchestrated coordination of several regulatory steps, which brings about subtle changes to the metabolic and physiological traits in ripening fruits. It is regulated by a complex network of transcription factors (TFs) and genetic regulators in response to endogenous hormones and external signals. With its progression, the color of fruits change owing to accumulation of pigments. Complex carbohydrates are converted to the sugars, the acidity of fruits decreases with the accumulation of sugars, the aroma compounds accumulate, and cell wall dynamics change, leading to softening [32].

ABA is a key factor in the regulation of fruit development and ripening among endogenous hormones, which is more important than ETH in the process of fruit ripening. It participates in the metabolism of many physiological active substances, such as glucose metabolism, pigment metabolism and aroma metabolism [4,10]. ABA can regulate the transport and distribution of assimilates; previous studies have demonstrated that ABA and sugar often have synchronous effects on diverse development processes, and the increase in ABA concentration is accompanied by an increase in sugar accumulation [33,34]. ABA accumulation occurs before ETH, and it has been shown that ABA promotes ripening by promoting ETH biosynthesis through up-regulation of ETH biosynthesis genes [35], so ABA has an automatic induction pathway of ETH during fruit ripening. In addition, ABA is a feedback regulator of pigment [12].

JAs are lipid-derived signaling molecules that control many developmental processes. JA can participate in pigment accumulation, cell wall modification, and ETH biosynthesis, and it can also interact with other hormones to participate in the regulation of fruit ripening [4,36]. Other studies have shown that JA treatment regulates aromatic substance content by regulating the LOX (lipoxygenase) pathway and the ETH biosynthesis and antioxidant system, and the synergistic regulation of ethylene biosynthesis and the ascorbate–glutathione cycle by methyl jasmonate contributes to the formation of aroma during the ripening of tomato fruits [37,38]; in addition, JA/ABA crosstalk can control plant metabolism and growth, and JA signal transduction is enhanced in presence of ABA [36].

Specifically, it shows that the phenotypic features of fruit development and ripening were regulated by molecular and genetic material. Recent discoveries have shed light on the molecular basis of developmental ripening control, suggesting common regulators of climacteric and non-climacteric ripening physiology [19,39,40]. Analyses of fruit-ripening mutants and ripening-related gene expression suggest higher levels of developmental regulatory cascades that remain to be defined [41]. 

In the present study, total RNA-Seq technologies were used to investigate the differences in the transcriptome and sRNAome between the early-ripening MT and its WT. We obtained many DEGs and differentially expressed miRNAs at different stages of fruit development and ripening, as shown in Figure 10.

### 4.1. Identification of Main Regulators and Metabolic Pathways Involved in Fruit Ripening

Plant hormones play key regulatory roles in fruit ripening, among which ABA plays a significant role in the fruit development and ripening of non-respiratory jumping fruits such as citrus [4,12]. In our study, the expression levels of *CsNCEDs* in MT fruit at the late development stage were significantly higher than those in WT fruit, so the ABA content accumulated in the peel of MT fruit by the late development stage was higher [11], as shown in Figure 4. In addition, the core conduction elements *PP2C* and *SnRK2* play an important role in regulating the ripening of citrus fruits in the ABA signal transduction pathway [42,43,44]. Due to the high accumulation of ABA in MT, ABA binds to the receptor (PYR/PYL/RCAR) and forms a complex with *PP2C* to inhibit the activity of *PP2C* phosphatase. Thus, *SnRK2* activity is released, and ABA signaling is transmitted downstream. Even though the expression level of *PP2C* in MT was higher than that in WT during fruit development, a large number of ABA-receptor binding compounds could also bind *PP2C* in MT. Additionally, the expression level of *SnRK2* in MT was higher than that in WT, so the function of *SnRK2* is upregulated in MT, and the ABA signal is smoothly transmitted downstream, leading to the discovery of early-ripening characteristics of fruits. This result is consistent with the findings of previous research [13,40].

JA also plays an important role in fruit ripening [32,36,45,46]. As shown in Figure 4, the expression levels of *OPR3* and *OPR2* [47,48] were higher in MT than in WT during fruit development, indicating that JA is formed in the peal earlier in MT than in WT [49]. Additionally, in the JA signal transduction pathway, the expression levels of JAZ proteins, which are the core negative regulatory factors, were higher in WT than MT during fruit development and the ripening stage, so the JA signal is smoothly transmitted downstream, which is another main reason for fruit showing early-ripening characteristics. The results are consistent with those of a previous study [45].

Sugar and acid metabolism play important roles in promoting fruit ripening [3,4,50]. As shown in Figure 4, α-glucosidase and β-fructofuranosidase, which are key enzymes in fructose synthesis in the starch and sugar metabolism pathways, had higher expression levels in MT than WT, and these two genes were found to be upregulated to promote fructose accumulation in our previous research. This result is consistent with the pattern of expression of sugar components in the early-ripening mutant of the pomelo variety [51]. *PG* and *PE* are key genes regulating fruit ripening and softening [52]. In MT, with the increases in *PG* and *GUAT* expression levels during fruit ripening, the expression level of *PE* was higher at the early stage of fruit development, and all the results indicated that pectin and cellulose were decomposed earlier and faster in MT fruits, so the characteristics of fruit ripening and softening appeared earlier [53]. In conclusion, the rapid accumulation of fructose and early decomposition of pectin and cellulose are important reasons for early maturity in MT fruits [54]. The appearance of easy peeling in MT fruit was also caused by the decline in the degree of adhesion of peel to flesh after the degradation of pectin and cellulose.

### 4.2. The Early-Ripening Trait Was Regulated by miRNAs in Citrus Fruit

sRNAs are involved in the regulation of plant growth, development and stress responses by silencing endogenous gene expression at either transcriptional or posttranscriptional levels [55]. miRNAs can be classified as conserved miRNAs or non-conserved miRNAs. Many miRNAs have been characterized in plants and play important roles in various signaling pathways [56]. For example, the target gene of miR156 is *SPL*, whose function is regulating development; the functions of miR159/*MYB* are signaling pathways and development; and miR172/*AP2* functions in signaling pathways, flower development, stress responses, etc. [57,58,59].

As an evolutionarily conserved microRNA, miR156 targets a subset of *SPL* genes in plants. *SPLs* are involved in a broad range of developmental processes in Arabidopsis, grape berry, the grass family Poaceae, sweet orange, and other plant species [60,61,62,63,64,65]. Through the functional analysis of four miR156 and 15 *SPL* genes that were retrieved from the *C**. sinensis* genome, Meiya Liu [61] found the ‘*miR156/CsSPL*’ module to be involved in starch accumulation. Mengjie Cui [62] found that treatment with an appropriate amount of exogenous hormones can promote or repress the grape berry ripening process, revealing that the ‘*Vv-miR156/VvSPL*’ pairs are related only to berry growth (IAA, GA, SA) or ripening (ABA, ETH and MeJA). In the present study, six ‘*miR156/CsSPLs*’ were identified, and the expression patterns of these genes were different between MT and WT during fruit development and ripening. These results indicate that miR156 may play an important role in citrus ripening.

MiR160, which targets auxin response factors (*ARFs*), mainly functions in plant growth, development, and stress responses [66,67,68]. Many studies have implicated hormone crosstalk in the fine-tuning of growth development processes, such as auxin–salicylic acid (SA) and auxin–jasmonic acid (JA) crosstalk [69]. Furthermore, Lin [70] found that during seed germination, communication between ABA and auxin by miR160-dependent reduction in auxin response factor 10 (*ARF10*) expression was observed [71]. Previous studies have shown that the *SlARF* family is involved in sugar metabolism during tomato fruit development [72], epidermal cell formation [73], and auxin and gibberellin signal transduction [74,75,76]. Zhang’s findings indicate that miR160a acts as a negative regulator inhibiting the synthesis of tanshinones by affecting the contents of JA, SA, and IAA in *S. miltiorrhiza* [66]. In the present study, the miR160 family and its targets were highly expressed and significantly different between MT and WT (Figure 8 and Figure 10) during fruit development and ripening. These results indicate that miR160 may be an important regulator and play a significant role in the formation of early maturity and easy-peeling traits in MT.

Laccases (LACs) are versatile enzymes that catalyze the oxidation of a wide range of substrates, thereby functioning in the regulation of plant developmental processes and stress responses [77,78], and *LACs* are widespread in plants [79,80,81,82]. In recent years, a miR397/*LAC* module was uncovered for domestication. Overexpression of miR397 has been found to interact separately with the laccase *OsLAC3* that has extensively improved rice yield by increasing grain size and promoting panicle branching in *japonica* rice by manipulating the brassinosteroid pathway [79], and miR397 has been shown to regulate fruit cell lignification in pear fruits by inhibiting the expression of the *LAC* [20]. In the present study, the expression of the miR397 family and its target *LACs* was significantly different between MT and WT (Figure 8 and Figure 10) during fruit development and ripening. These results indicate that miR397 may be an important regulator of the fruit quality trait in MT.

The targets of miR3954 (*NAM*, *ATAF1–2*, and *CUC2*) are all *NACs*, have the same RNA-binding domain, and mainly function in plant flower and fruit development and stress responses [83,84,85,86]. *NAC* TFs form a plant-specific superfamily, and different members perform a variety of important functions; for example, the two *NAC* family genes have been shown to be involved in the regulation of fruit ripening (including fruit softening, color change, ethylene biosynthesis, etc.) [87] and be associated with abiotic stress (drought, cold stress, etc.) [88]. In our study, the expression of the miR3954 family and its target *NACs* was significantly different between MT and WT plants (Figure 8 and Figure 10) during fruit development and ripening. These results indicate that miR3954 may be an important regulator of fruit development and stress in MT.

## 5. Conclusions

The results demonstrated that there are several key candidate regulators that play important roles in citrus fruit development and ripening. Comparative transcriptomic and sRNAome analyses identified that some biological processes were crucial, such as starch and sucrose metabolism, the plant hormone signal transduction pathway, and the carotenoid biosynthesis pathway. Additionally, miRNA/target gene pairs played major roles in affecting citrus fruit ripening and influenced fresh quality characteristics, such as softening and sugar content, etc. These results have provided new insights into the molecular mechanisms underlying citrus ripening and quality regulatory networks.

## Figures and Tables

**Figure 1 genes-13-01706-f001:**
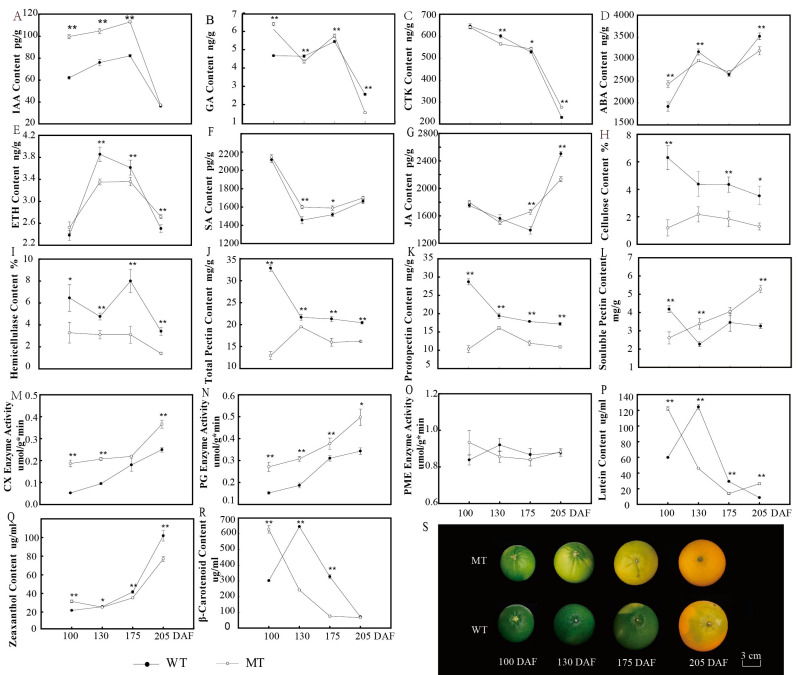
The trend of metabolites at different stages of fruit development in MT and WT. (**A**–**G**), Content of endogenous hormones during the four development and ripening stage, 100, 130 175 and 205 days after flowering, respectively. (**H**–**L**), Content of the peel structural material. (**M**–**O**), Activity of the related enzymes of peel structural material. (**P**–**R**), Content of pigment accumulation in citrus peel. (**S**), Different development and ripening stages of MT and WT. * and ** indicate significant differences at *p* < 0.05 and *p* < 0.01 levels, respectively, between WT and MT at a particular ripening stage based on Student’s *t*-test.

**Figure 2 genes-13-01706-f002:**
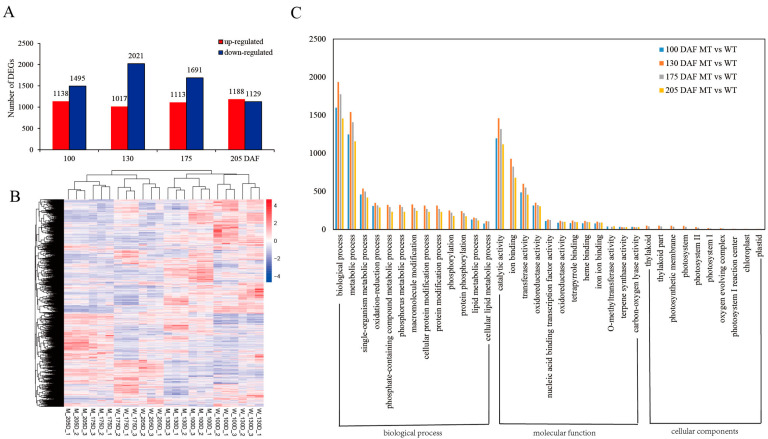
Analysis of transcriptome sequencing. (**A**) Numbers of DEGs between MT and WT at 100, 130, 175, and 205 DAF. (**B**) Cluster heatmap based on different samples of different genes. Red represents up-regulation, and blue represents downregulation. (**C**) GO Functional classification in MT and WT.

**Figure 3 genes-13-01706-f003:**
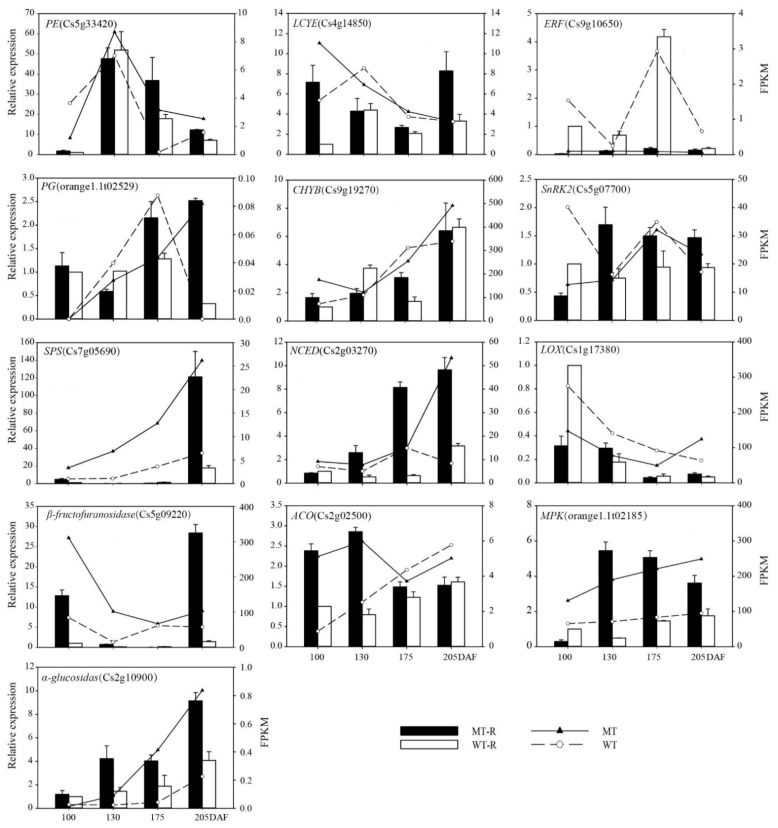
qRT-PCR validation of DEGs. A total of 13 genes, which involved in starch and sucrose metabolism, carotenoid biosynthesis, and plant hormone synthesis and signal transduction with the fruit development and ripening. MT-R (black bar) and WT-R (white bar) represented the data by qRT-PCR; MT (solid line) and WT (broken line) represented the data by RNA-Seq.

**Figure 4 genes-13-01706-f004:**
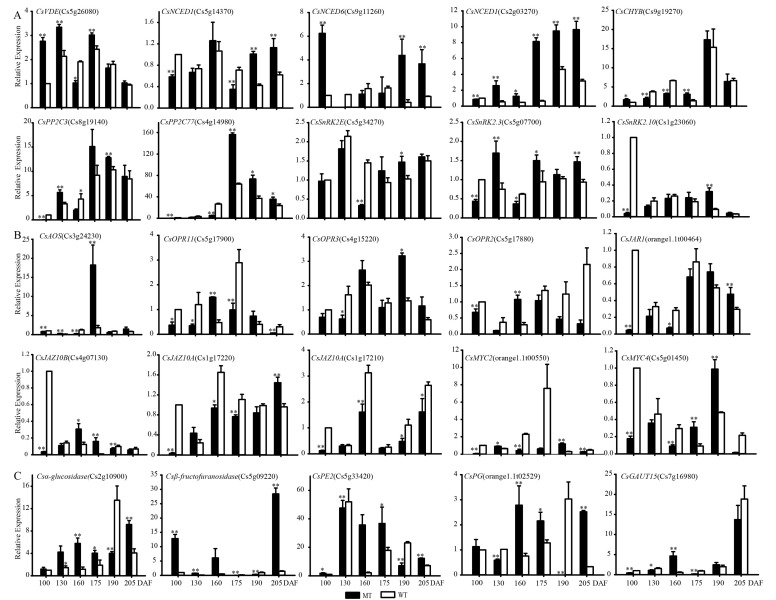
Transcriptional level of genes in ABA synthesis and signal transduction pathway (**A**), JA synthesis and signal transduction pathway (**B**), and a part of starch and sucrose metabolism pathway (**C**). * and ** indicate significant differences at *p* < 0.05 and *p* < 0.01 levels, respectively, between WT and MT at a particular ripening stage based on Student’s *t*-test.

**Figure 5 genes-13-01706-f005:**
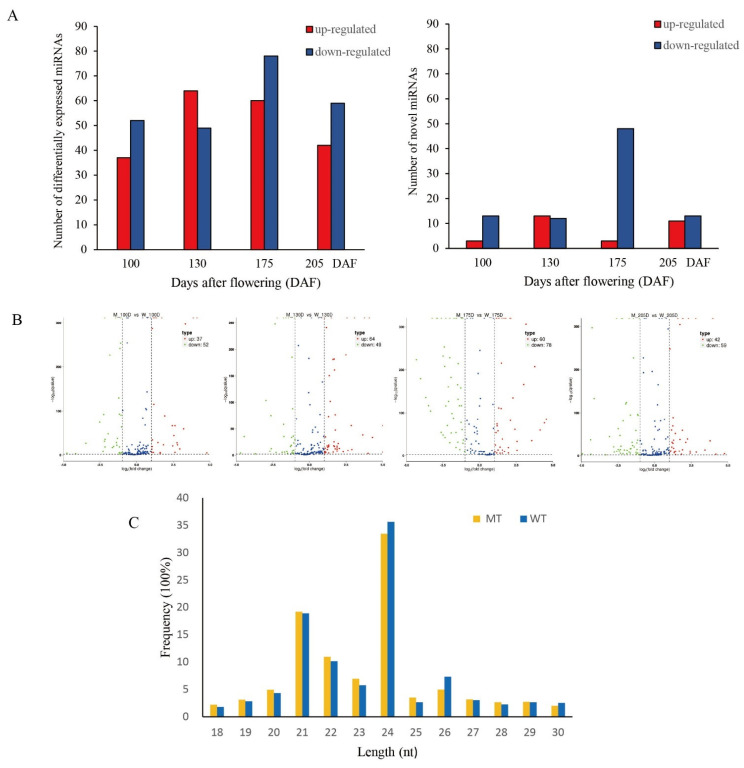
Overview and analysis of miRNAs sequencing. (**A**) Number of differentially expressed miRNAs between MT and WT at 100, 130, 175, and 205 DAF. (**B**) The volcano map based on different samples of different miRNAs. Red represents up-regulation, and green represents downregulation. (**C**) Length distribution of the sRNA reads in MT and WT.

**Figure 6 genes-13-01706-f006:**
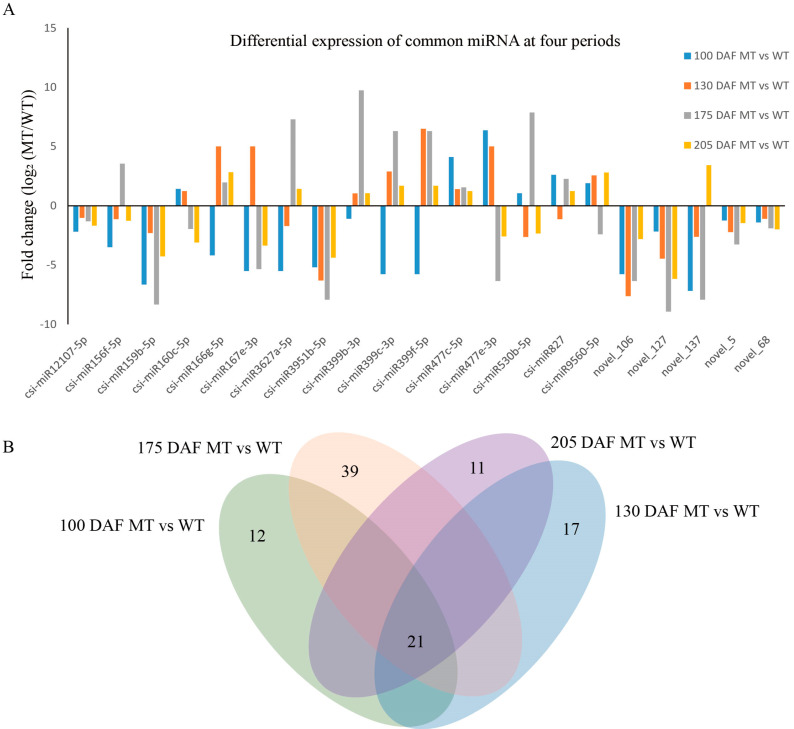
Statistical map of differential expressed miRNA. (**A**) The differentially expressed of major known miRNAs and novel miRNAs between MT and WT, the fold change is shown as a log_2_ ratio. (**B**) Venn diagram of shared differential expressed miRNAs between the four stages.

**Figure 7 genes-13-01706-f007:**
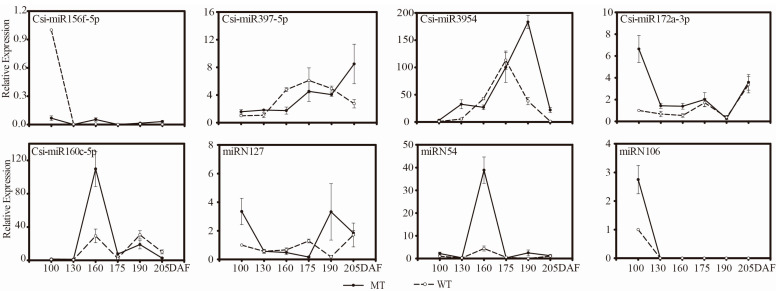
Expression patterns of major miRNAs involved in fruit development and ripening.

**Figure 8 genes-13-01706-f008:**
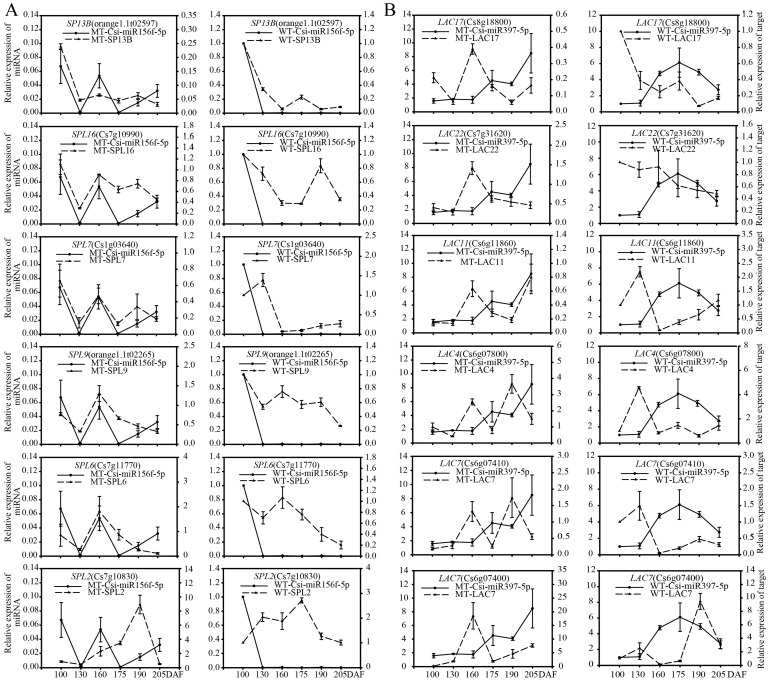
Expression patterns of known miRNAs and their target genes in MT and WT involved in fruit development and ripening. (**A**), csi-miR156f; (**B**), csi-miR397; (**C**), csi-miR172; (**D**), csi-miR160; (**E**), csi-miR3954.

**Figure 9 genes-13-01706-f009:**
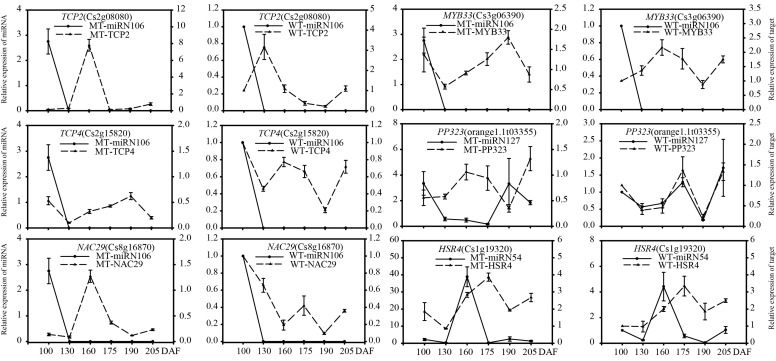
Expression patterns of novel miRNAs and their target genes in MT and WT involved in fruit development and ripening. miRN106/*TCP2*, miRN106/*TCP4,* miRN106/*NAC29,* miRN106/*MYB33,* miRN127/*PP323*, miRN54/*HSR4*.

**Figure 10 genes-13-01706-f010:**
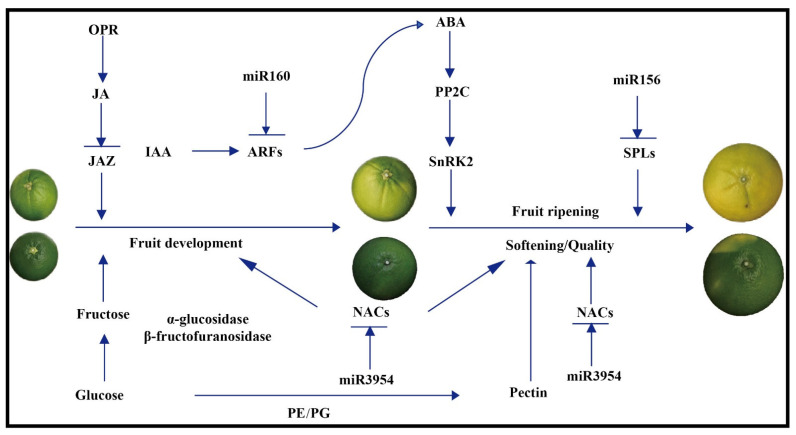
A schematic model with the proposed roles of transcription factors and miRNAs involved in citrus fruit development and ripening.

**Table 1 genes-13-01706-t001:** Major KEGG pathways related to metabolism in MT compared with WT at 100, 130,175, and 205 DAF.

KEGG Pathway	Gene Numbers			
	100 DAF	130 DAF	175 DAF	205 DAF
Metabolic pathways	252	301	273	237
Biosynthesis of secondary metabolites	169	191	157	157
Plant–pathogen interaction	48	34	46	30
Plant hormone signal transduction	41	25	43	29
Protein processing in endoplasmic reticulum	34	32	15	17
Starch and sucrose metabolism	34	28	32	29
Phenylpropanoid biosynthesis	31	32	25	35
Biosynthesis of amino acids	31	34	35	24
Carbon metabolism	31	44	40	18
Glycolysis/Gluconeogenesis	17	26	15	14
Terpenoid backbone biosynthesis	17	14	15	10
Sesquiterpenoid and triterpenoid biosynthesis	15	19	16	12
Peroxisome	14	15	15	13
Fatty acid metabolism	13	11	12	14
Pentose and glucuronate interconversions	13	15	15	11
Fatty acid degradation	11	15	9	12
Glyoxylate and dicarboxylate metabolism	10	19	17	9
Ascorbate and aldarate metabolism	10	11	6	5
Biosynthesis of unsaturated fatty acids	9	5	6	6
Carotenoid biosynthesis	9	8	8	6
Tropane, piperidine and pyridine alkaloid biosynthesis	9	6	5	7
Ubiquitin mediated proteolysis	9	12	7	9
Photosynthesis	8	35	25	12
Nitrogen metabolism	8	10	8	7
Fatty acid biosynthesis	7	7	9	11
Degradation of aromatic compounds	6	8	5	5
Cutin, suberine and wax biosynthesis	6	8	6	9
Flavonoid biosynthesis	5	12	9	25

**Table 2 genes-13-01706-t002:** A list of the important differentially expressed genes in MT and WT during the fruit development and ripening; these genes are involved in starch and sucrose metabolism, carotenoid biosynthesis and plant hormone signal transduction; the fold change of 100, 130, 175, and 205 DAF represent the MT compared with WT. *GUAT*, α-1,4-galacturonosyltransferase; *SS*, sucrose synthase; *SPS*, sucrose-phosphate synthase; *PE*, pectinesterase; *PG*, polygalacturonase; *PSY*, phytoene synthase; *ZDS*, zeta-carotene dehydrogenase; *LCYE,* ε-lycopene cyclase; *CHYB*, β-carotenoid hydroxylase; *NCED*, 9-cis-epoxycarotenoid dioxygenase; *SnRK2,* serine/threonine-protein kinase SRK2; PP2C, protein phosphatases 2C; *ACS*, ACC synthase; *ACO*, ACC oxidase; ETR, ethylene receptor; *MPK*, mitogen-activated protein kinase; *EBF1*, EIN3-binding F-box protein1; AHP, histidine phosphotransfer proteins; *B-ARR*, Arabidopsis type-B response regulators; *AUX1*, auxin resistant1; *JAR1*, asmonic acid—amido synthetase 1; JAZ, Jasmonate ZIM-domain; *MYC2*, myelocytomatiosis proteins2; *TGA*, TGACG motif-binding factor; *LOX*, lipoxygenase; *AOS*, allene oxide synthase; *OPR*, 12-oxo-phytodienoic acid reductase.

Gene Id	Gene Description	Log2ratio (MT/WT)			
		100 DAF	130 DAF	175 DAF	205 DAF
Starch and sucrose metabolism					
Cs7g16980	*GAUT*	−1.71	−1.37	-	−1.17
Cs6g10710	*GAUT*	−0.98	-	-	-
Cs4g06850	*SS*	−1.11	-	-	−1.16
Cs6g14340	β-fructofuranosidase	−3.60	−3.04	-	−3.12
Cs5g09220	β-fructofuranosidase	1.65	2.66	-	-
Cs7g05690	*SPS*	1.53	2.57	1.69	1.96
orange1.1t03668	*SPS*	1.09	0.66	0.75	0.67
Cs4g05380	*SPS*	0.96	1.04	-	-
Cs5g19060	*SPS*	1.57	-	−2.90	−1.88
Cs2g10900	α-glucosidase	−0.77	-	3.04	1.86
Cs4g06710	*PE*	−1.07	−1.82	−1.29	−1.55
Cs5g33420	*PE*	−1.84	-	3.97	-
Cs2g17070	*PE*	1.71	-	-	−2.97
orange1.1t02530	*PG*	-	-	-	6.66
orange1.1t02529	*PG*	-	-	-	6.22
Carotenoid biosynthesis					
orange1.1t02108	*PSY*	−1.69	-	−1.77	−3.48
Cs6g15910	*PSY*	-	−0.84	−1.16	-
Cs3g11040	*ZDS*	1.17	-	-	-
Cs4g14850	*LCYE*	0.83	-	-	-
Cs9g19270	*CHYB*	1.05	-	-	-
Cs5g14370	*NCED*	−1.63	−1.86	−2.07	-
Cs9g11260	*NCED*	-	-	−4.52	-
Cs2g03270	*NCED*	-	-	-	2.58
Plant hormone signal transduction					
Cs5g34270	*SnRK2*	−0.69	-	-	-
Cs5g07700	*SnRK2*	−1.80	-	-	-
Cs1g23060	*SnRK2*	−0.78	−1.47	-	−1.51
Cs8g19140	*PP2C*	1.09	-	-	-
Cs4g14980	*PP2C*	−1.28	-	-	-
Cs5g03060	*ACS*	-	−1.50	−0.93	-
orange1.1t00416	*ACS*	−3.32	-	−2.86	-
Cs2g02500	*ACO*	2.31	-	-	-
Cs5g20590	*ACO*	-	−0.91	-	-
Cs9g08850	*ETR*	−1.70	-	−0.84	-
orange1.1t02185	*MPK*	0.77	1.34	1.28	1.35
Cs4g17960	EBF1	−2.08	-	-	-
Cs5g29870	*ERF1*	−3.32	-	−4.13	-
Cs9g10650	*ERF1*	−4.24	-	−5.21	−3.53
orange1.1t01539	AHP	1.90	3.58	1.74	1.05
Cs9g02730	*B-ARR*	0.69	-	-	-
Cs4g01640	*B-ARR*	0.84	-	-	-
Cs8g02900	*AUX1*	2.26	-	-	−7.32
Cs7g31320	*AUX1*	1.32	-	-	-
orange1.1t00464	*JAR1*	−1.42	−0.95	−1.35	-
Cs1g17220	*JAZ*	−1.15	-	−1.28	-
Cs4g06520	*JAZ*	−0.82	-	−0.81	-
Cs7g02820	*JAZ*	−2.21	-	-	-
Cs4g07130	*JAZ*	−0.91	-	−1.14	-
Cs1g17210	*JAZ*	−1.53	-	−1.00	-
orange1.1t00550	*MYC2*	−2.37	-	−2.00	-
orange1.1t01021	*MYC2*	−0.85	−0.83	−1.23	-
Cs5g01450	*MYC2*	−0.94	-	−0.93	-
Cs7g19070	*TGA*	7.07	3.70	-	−2.26
orange1.1t03769	*LOX*	6.68	-	-	-
orange1.1t04376	*LOX*	−2.12	−1.13	−0.92	-
Cs1g17380	*LOX*	−1.14	−0.90	−1.06	-
orange1.1t03771	*LOX*	−7.34	-	-	-
Cs3g24230	*AOS*	−1.46	-	-	-
orange1.1t03727	*OPR*	−1.33	-	-	-
Cs5g17900	*OPR*	−1.47	-	−2.74	-
Cs4g15220	*OPR*	−1.02	-	-	0.77
Cs5g17880	*OPR*	−1.21	-	−2.73	-

**Table 3 genes-13-01706-t003:** Statistics of sRNA at different stages of fruit development in MT and WT.

Sample	Clean Reads	Total Reads (18–30 nt)	Unique Reads	Mapped sRNA	Known-miRNA	Novel-miRNA
M_100D	14,316,231	4,176,010	1,518,461	3,246,961 (77.75%)	65,163 (2.01%)	3554 (0.11%)
M_130D	15,465,813	5,296,231	1,879,119	4,146,086 (78.28%)	114,988 (2.77%)	5976 (0.14%)
M_175D	16,418,922	1,906,354	818,850	1,385,348 (72.67%)	29,872 (2.16%)	1019 (0.07%)
M_205D	15,237,868	8,968,818	2,202,926	6,648,319 (74.13%)	196,588 (2.96%)	11,140 (0.17%)
W_100D	16,423,940	8,762,671	2,381,531	6,451,132 (73.62%)	236,285 (3.66%)	11,183 (0.17%)
W_130D	16,986,556	6,362,575	1,736,884	5,148,648 (80.92%)	164,510 (3.20%)	7221 (0.14%)
W_175D	14,755,090	3,676,819	1,068,268	2,970,459 (80.79%)	39,118 (1.32%)	4677 (0.16%)
W_205D	16,507,044	7,430,258	1,729,613	5,695,338 (76.65%	225,001 (3.95%)	12,835 (0.23%)

**Table 4 genes-13-01706-t004:** Target genes of major miRNAs involved in fruit development and ripening. *SPL*, SQUMOSA PROMOTER BINDING PROTEIN-LIKE; *ARF,* auxin response factor; *LAC*, laccase; *AP2*, APETALA2; *NAC* (no apical meristem (NAM), Arabidopsis transcription activation factor (ATAF1/2), cup-shaped cotyledon (CUC2)); *TCP*, teosinte branched1/cincinnata/proliferating cell factor; *PP323*, protein phosphatase323; *HSR*, heat-shock response factor.

miRNA/Target Gene	Target-Start	Target-End	Score	log2 (MT/WT)				Target Annotation
				100 DAF	130 DAF	175 DAF	205 DAF	
csi-miR156f-5p				−3.48	−1.11	3.55	−1.25	
orange1.1t02597	1203	1223	0.5					SP13B
Cs7g10990	1536	1556	0.5					SPL16
Cs1g03640	1585	1605	0.5					SPL7
Cs2g05730	807	827	1.5					SPB1
orange1.1t02265	996	1016	1.5					SPL9
Cs2g23550	892	912	1.5					SPB2
Cs7g11770	1477	1497	1.5					SPL6
Cs7g10830	1172	1192	1.5					SPL2
csi-miR160c-5p				1.43	1.24	−1.94	−3.09	
Cs8g16440	1967	1987	0.5					ARFR
Cs7g25670	1730	1750	0.5					ARFR
Cs6g11800	2142	2162	2					ARFR
Cs3g18940	1412	1432	2					ARFQ
csi-miR397-5p				1.61	–	7.88	3.8	
Cs6g11860	736	756	1.5					LAC11
Cs6g07800	734	754	1.5					LAC4
Cs8g19850	669	689	2					LAC4
Cs6g07410	764	784	2					LAC7
Cs7g23490	705	725	2.5					LAC17
Cs6g07400	789	809	2.5					LAC7
Cs6g07450	878	898	2.5					LAC7
Cs8g18800	711	731	2.5					LAC17
Cs6g06920	693	713	3					LAC17
Cs6g06880	713	733	3					LAC17
Cs8g17630	724	744	3					LAC17
Cs8g17350	690	710	3.5					LAC17
Cs6g06890	789	809	3.5					LAC17
Cs7g31620	685	705	3.5					LAC22
csi-miR172a-3p				–	−2.69	7.3	–	
Cs7g27790	1604	1623	1					RAP27
Cs6g04120	1535	1554	2					AP2
orange1.1t04055	2039	2058	2					AP2
Cs8g17390	1810	1829	2					RAP27
csi-miR3954				–	–	1.25	1.04	
orange1.1t05093	75	95	2					NAC40
Cs1g09660	148	168	2					NAC40
orange1.1t05587	237	257	2.5					NAC79
Cs1g09710	156	176	3					NAC40
Cs7g22510	75	95	4					NAC4
Cs7g22470	178	198	4					NAC4
Cs7g22540	75	95	4					NAC91
miRN106				−5.76	−7.61	−6.33	−2.79	
Cs2g08080	2017	2037	3.5					TCP2
Cs3g06390	762	782	4					MYB33
Cs8g16870	204	224	4.5					NAC29
Cs2g15820	1428	1448	4.5					TCP4
Cs7g25460	2050	2070	4.5					TCP4
miRN127				−2.16	−4.45	−8.91	−6.15	
orange1.1t03355	751	773	5					PP323
miRN54				–	–	−8.65	−1.05	
Cs1g19320	1097	1120	5					HSR4

## Data Availability

Not applicable.

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
