# Peer review of "Comparative Transcriptome and sRNAome Analyses Reveal the Regulatory Mechanisms of Fruit Ripening in a Spontaneous Early-Ripening Navel Orange Mutant and Its Wild Type"

_genes, 2022, doi:10.3390/genes13101706_

Round 1

Reviewer 1 Report

Dear Authors,

I find the manuscript entitled "Comparative transcriptome and sRNAome analyses reveal the regulatory mechanisms of fruit ripening in a spontaneous early-ripening navel orange mutant and its wild type" interesting however it needs several improvements. 

I posted my comments below as they appear in the text.

Latin names when first used should be full i.e. three-element. 

Line 47-48: This sentence should be moved to paragraph line 84-93. Here, it is misleading and implies that NCED1 is a transcription factor. What is missing here is the citation, the full name of the gene and the information that it is a key gene in ABA biosynthesis. 

Line 91-93: Instead of FAKAT1, it should be FaKAT1, indicating that it is a gene derived from strawberry. The name of this gene should be written in italics. The entire manuscript should be reviewed for this, as there are more errors of this type. 

The notation miR156-SPL should be changed to miR156/SPL. The hyphen is misleading in suggesting that SPL is part of the miR name, not the target gene. Italics should also be used here.

Line 102: "n" ?

Line 116 "d"?

The description of the method lacks information on how many biological replicates were performed and how many plants they consisted of. 

Also missing is information on what number of reads were assumed in the sequencing analysis.

The description of the data analysis is too laconic. Information about the bioinformatics pipeline is missing. Instead, information about the programs used appears quite often in the description of the results. In line 180, it is not clear what Q20 and what Q30 refer to. Were the results mapped to the reference genome? If so, it should be indicated. 

The figures in the entire manuscript need to be improved. Including a million of small diagrams misses the point. You need to select the ones that are most important and move the rest to the supplement. The coloring of the graphs also needs to be corrected because the color differences are too small and the figures will be completely unreadable to people with impaired color vision (Fig2c, 3, 4, 5c)

Tables look like pasted images, this with low resolution. Perhaps this is just a problem with converting to pdf. 

The discussion also needs to be refined. The results obtained need to be placed in the broad context of fruit ripening physiology. In the case of ABA and JA, it is necessary to state what role these hormones play in fruit ripening. 

Best regards,

M.

Author Response

Dear reviewer,

Thanks very much for your time to review this manuscript. I really appreciate all your comments and suggestions. We have considered these comments carefully and triedour best to address every one of them.

Reviewer 2 Report

Comments to authors

The work by Mi et al. advances our understating the roles of sRNAome in the development and ripening of orange fruits. The report is well written and condensed, as well as technically appropriate for Genes-MDPI.  However, before being able to recommend acceptance, I invite authors to address the following amendments.

1-     In line 12: "(‘Ganqi 4’, C. sinensis L. Osbeck)"Please replace "C. sinensis" by the full scientific name of Citrus sinensis.

2-     In line 24:  "csi-miR3954" it is repeated twice in the sentence.

3-     In line 66 and 69: Please replace the name of "ζ-carotene" by "zeta-carotene".

4-     In line 15, 72, 144, 344, 349 (Figure 3), 501 and 540: “RNA-seq”,   it should be “RNA-Seq”.

5-     In line 47: Please add the full name of “NCED1” before the abbreviation; it should be "Nine-cis-epoxycarotenoid dioxygenase (NCED1).

6-     In line 55, 59, 83: Please add the full name of these genes “PSY, SAM, ACC, LCYB1, PDS and CCD1” before the abbreviation, for example PSY it should be Phytoene synthase (PSY) gene.

7-     In line 57: "The MADS box family transcription factors" it should be "The MADS-box family of transcription factors".

8-     In line 85: Please reduce the space between the pronoun "These" and the dot.

9-     In line 102: What is this meaning "n" ?

10-  In line 113:  Please include information about the geography GPS location of this area "The WT ‘Newhall’ navel orange (C. sinensis L. Osbeck) and its spontaneous earlyripening mutant (MT) ‘Ganqi 4’ were both cultivated in the same orchard (Longnan, Gan-113 zhou City, China)."

11-  In line 128: "Each sample had three repetitions" it should be "Each sample has three repetitions".

12-  In line 136:  "PG and CX enzymes" it should be polygalactouronase (PG) and encellulase(Cx).

13-  In line 138 - 141: Please add more details about this method and if this method for determine the endogenous hormone by enzyme-linked immunoassay has been used before in other research articles (Add the references). On the other hand, this Company (Shanghai Heng Yuan 140 Biological Technology Company) has many kits products (http://en.hnybio.com/plant/321b5572.shtml ), so please add the full name of this kit.

14-  In line 114-116 and 130-131: In materials and research methods, you mentioned that the samples were taken at the following times (90, 120, 180 and 210 DAF). But in the results it was mentioned that the samples were taken at the following times (100, 130, 175 and 205 DAF). Please explain this, and which times are correct.

15-  In line 241: "The content in MT was significantly higher than that in WT before fruit degreening" this sentence with no meaning it should be the content of .................in MT........

16-  In line 272: Please add the second bracket for Fig. 2B.

17-  In line 283: Please replace "GOseq software" by "goseq package".

18-  In line 316: In sometimes you used accession number in the form (e.g. Cs2g02500) and in other position you used in form (e.g. orange1.1t00416), what is the different between these two accessions.

19-  Please add your tables in your manuscript in word table text format not as figures.

20-  In line 356:  Please add the full name of these genes "CsVDE, CsCHYB, PP2C, GUAT and SnRK2".

21-  The name of all genes in your manuscript should be italic such as, SnRK2, PP2C, OsLAC3, OPR3 and OPR2.

22-  In line 260, 263 and 264: You should add the abbreviation name of the enzyme only, because we add the full name for abbreviation at first time (e.g. Cx, PG and PME).

23-  In line 593: Please reduce the space between the pronoun "are" and "signaling".

24-  In line 433: "(Fig. S1-S2)," it should be "Fig. S1 and S2".

25-  In line 503: Please replace "215 DAF" by "205 DAF".

26-  In line 564: Figure 5 did not show the expression levels of OPR3 and OPR2, Please check the figure number carefully.

27-  In line 598: "Arabidopsis" it should be not italic, because it is a common name of Arabidopsis plant.

28-  In some position you say "at different stages of fruit development" and in other position you used "at four periods of fruit development", please use only one in the entire manuscript.

29-  Some figures, such as Figure 1, 4, 7 and 8 needs improvements.

30-  In line 600: Please use the abbreviation for the scientific name of Citrus sinensis. (Note; we used the full scientific name only at the beginning of the manuscript, after that we used only the abbreviation).

31-  In figure 5 A: please add the number of differentially expressed miRNAs and  number of novel miRNAs in " y " chart axis, and the  days after flowering (DAF) in the " x " chart axis.

32-  Based on the materials and methods, you mentioned that you collected the samples from WT and MT at four different periods or stages for RNA-Seq, which are “90, 120, 180 and 210” While data in the results, discussion and figures 4, 7, and 8, such as (Transcriptional level and Expression patterns) at six different periods or stages "100, 130, 160, 175, 190 and 205", please explain.

33-  Please adjust the format and style of the figure 8, and in the case if the images and data are so many, the results in the figure can be divided into several figures, which will contribute to displaying the data in a clearer form.

34-  In figure 9: There is a dash (˞) between PG and PE, Softening and Quality. What is this meaning?

35-  In line 566: Please replace the name of “jasmonic acid” by abbreviation “JA”.

36-  In Table 2: Please add a suitable gene description, and add this sentence "These genes involved in starch and sucrose metabolism, carotenoid biosynthesis and plant hormone signal transduction, the fold change of 100, 130,175, and 210DAF represent the MT compared with WT"  as figure legend.

Author Response

(The authors gave the same response as above.)

Round 2

Reviewer 1 Report

The provided version of the manuscript is not in change tracking mode, which makes the work significantly more difficult.

The miRNA-target notation remains in many places. 

In the description of the methodology, please remove the sections that overlap with the kit manufacturer's protocol you refer to. They only take up space. 

In the description of the results,RSEM and Cuffdiff programs appear, which have no citations and are not included in the methodology in the results analysis section. 

I hadn't noticed this before, but there is no information on how the sRNA was analyzed. To which database and its version the sequenced miRNAs were compared.

Figure 8 is still unreadable. The individual graphs are too small to read anything from them. To make sense they must be at least the size of those in Figure 7. 

The discussion has been improved, but it is still quite shallow and should be better grounded in physiology. 

Author Response

Dear reviewer,

Thanks very much for your time to review this manuscript again. I really appreciate all your comments and suggestions. We have considered these comments carefully and triedour best to address every one of them.
